# Temperature Sensor Based on Periodically Tapered Optical Fibers

**DOI:** 10.3390/s21248358

**Published:** 2021-12-14

**Authors:** Bartlomiej Guzowski, Mateusz Łakomski

**Affiliations:** Department of Semiconductor and Optoelectronic Devices, Lodz University of Technology, 93-590 Lodz, Poland; mateusz.lakomski@p.lodz.pl

**Keywords:** optical fiber, optical sensor, tapered optical fiber, temperature sensor

## Abstract

In this paper, the fabrication and characterization of a temperature sensor based on periodically tapered optical fibers (PTOF) are presented. The relation between the geometry of the sensors and sensing ability was investigated in order to find the relatively simple structure of a sensor. Four types of PTOF structures with two, four, six and eight waists were manufactured with the fusion splicer. For each PTOF type, the theoretical free spectral range (FSR) was calculated and compared with measurements. The experiments were conducted for a temperature range of 20–70 °C. The results proved that the number of the tapered regions in PTOF is crucial, because some of the investigated structures did not exhibit the temperature response. The interference occurring inside the structures with two and four waists was found be too weak and, therefore, the transmission dip was hardly visible. We proved that sensors with a low number of tapered regions cannot be considered as a temperature sensor. Sufficiently more valuable results were obtained for the last two types of PTOF, where the sensor’s sensitivity was equal to 0.07 dB/°C with an excellent linear fitting (*R*^2^ > 0.99). The transmission dip shift can be described by a linear function (*R*^2^ > 0.97) with a slope α > 0.39 nm/°C.

## 1. Introduction

Temperature measurement can be performed using one of three methods: the non-electric method, the pyrometric method and the electric method. The non-electrical method uses a change in the physical state or physical or chemical parameters of the object. In the pyrometric method, the thermal radiation of the object is investigated. The third method is based on the relationship between the change in object temperature and resistance (resistance sensors) or the relationship between the thermoelectric strength of a thermocouple and temperature (generation sensors). There are a large number of different electric temperature sensors, e.g., negative temperature coefficient (NTC) thermistors [1], resistance temperature detectors, thermocouples [2] or semiconductor temperature sensors [3,4]. Recently, optical fiber sensors have been intensively developed for a large number of applications, such as: temperature measurement [5,6,7,8], displacement sensing [9,10,11,12], strain/pressure detection [13,14,15,16,17], and bio-photonic and medical tests [18,19,20]. Due to their dielectric nature, numerous advantages can be distinguished: contactless operation, small dimension and weight, high efficiency and low cost. Different types of fiber optic temperature sensors have been proposed, e.g., fiber Bragg gratings (FBG) [5,21,22], tapered optical fibers [6,23,24], long-period gratings (LPG) [7,25,26], and Fabry–Perot (F-P) [27,28,29] or modal interferometers [8,11]. The sensitivity of temperature sensors based on FBG is limited due to small thermo-optic coefficient and glass thermal expansion [20,30]. The sensitivity of sensors based on F-P interferometers can be very high; for example, in [29] it was >2.70 nm/°C, but only in the specific range of 51.2–70.5 °C. For high-temperature applications, multicore fibers have been proposed [31]. These structures enable measurement of temperature up to 1000 °C with sensitivity of ~36.8 pm/°C. For higher temperature ranges, sensors based on sapphire fiber have been developed [32,33,34,35,36] and their sensitivity did not exceed 35.7 nm/°C. The combination of sapphire fiber and FBG allows even 1900 °C to be investigated [35]. PMMA optical fibers were used to fabricate sensors for lower temperature applications (<110 °C) [37]. A very complex structure of a few tapered mode fibers with LPG was also investigated as a temperature sensor [38]. The structure reached the sensitivity of 39.3 pm/°C in the range of 30–90 °C. Among these sensor types, fiber optic Mach–Zehnder interferometers (MZI) are attractive for researchers because they are compact and relatively low cost [24,39,40,41,42,43]. In [44], using a cascaded sensor, where a capillary hollow-core fiber was placed between two sections of multimode fibers, temperature sensitivity reached 1.964 nm/°C in a range from 10 to 70 °C. In [45], a sensor with the cascaded configuration of MZI and F-B was presented. The impressive temperature sensitivity was equal to 6.82 nm/°C in a range from 10 to 60 °C. Another cascade of single-mode fiber, multimode fiber (MMF) and dual core fiber placed between MMFs was investigated by Zhao et al. [46]. In this case, the sensor achieved sensitivity of 2.18 nm/°C, although in a narrow range of temperature of 26–44 °C. The MZI can be utilized in many other applications, e.g., as a refractive index sensor [47,48,49] or a strain sensor [50,51].

Our goal was to develop a fiber optic temperature sensor based on periodically tapered optical fibers (PTOF) with a simple construction, low-cost production and, simultaneously, good sensing abilities. Therefore, in this study, we investigated the relation between the number of tapered regions in the developed PTOF and their sensing performance, in order to find the simplest sensor construction with good sensitivity. Four models of PTOF sensors, having 2, 4, 6 and 8 waists, were manufactured, and detailed information related to the principle of operation, and fabrication and characterization processes, is given. The expected free spectral ranges (FSR) between the interferometric fringes were calculated and confirmed by experimental results. The obtained results show that only two types of fabricated structures can be used as temperature sensors. The sensitivity of the developed sensors was compared with similar existing sensors and possible applications are given.

## 2. Principle of Operation

In a PTOF with a period of tapering *L_p_*, four regions can be distinguished, as shown in Figure 1. The tapered-down and tapered-up regions are labeled in Figure 1 as region *l*_d_ and *l*_u_, respectively. The waist region called *l* is placed between the *l*_d_ and *l*_u_ regions. The fourth element of the PTOF is a section of non-tapered fiber *l*_s_. The geometry of the sensor, particularly the symmetry and uniform waist region diameter *d*, are the key parameters of the PTOF. In our previous work [52], we showed that we were able to fabricate tapered fibers with the desired geometry in a repeatable manner. Based on this experience, we developed the fabrication process of PTOF structures. 

The sensor principle of operation is based on MZI. Some of the light injected into the PTOF leaks from the core into a cladding in the taper-down region. Therefore, higher order modes are excited in the cladding area. By comparison, the cladding modes in the taper-up region are coupled back to the core region. Due to the phase difference ∆*ϕ* between the cladding and the core modes, the Mach–Zehnder interferometer is created in the sensor [50,53]. Equation (1) describes the relative phase difference between these interfering modes:(1)$$\Delta \phi =\frac{2\pi }{\lambda }\Delta n_{eff}L$$
where ∆*n_eff_* is the difference between core and cladding effective refractive indices, *L* is the interferometric length, and *λ* is the central wavelength of light. Transmission dip appears for the phase difference equal to ∆*ϕ* = (2*k* + 1)*π*, where *k* is an integer. The transmission light intensity *I* is described with Equation (2) [53,54,55], and it will change with the change in temperature, causing variation in output power:(2)$$I=I_{co}+I_{cl}+2\sqrt{I_{co}+I_{cl}}cos\phi$$
where *I_co_* and *I_cl_* are intensities of the core and cladding modes, respectively.

The distance between the transmission dips is given by Equation (3) [56,57]:(3)$$\Delta \lambda_{m}\approx \frac{\lambda^{2}}{\Delta n_{eff}L}$$

The temperature variation causes the modification of the refractive index of the core and cladding [58,59,60] due to thermo-optic effects, in addition to the variation of *L* due to glass thermal expansion. Therefore, the temperature sensitivity of the sensor can be described by Equation (4) [61]:(4)$$\frac{d\lambda }{dT}≅\frac{\frac{2L}{2k+1}\left(\frac{\partial \Delta n_{eff\;}dn_{co}}{dT\;\partial n_{co}}+\frac{\partial \Delta n_{eff\;}dn_{cl}}{dT\;\partial n_{cl}}\right)+\frac{\lambda }{L}\;\frac{dL}{dT}}{1-\frac{2L}{2k+1}\;\frac{\partial n_{eff\;}}{\partial \lambda }}$$

## 3. Fabrication Process

The Furukawa Fitel S153A optical fusion splicer and single mode G.652.B optical fiber (SMF) were used to fabricate the PTOF sensors. The SMF (1) was placed in V-grooves (4) and it was held by the fusion splicer holders (3), as presented in Figure 2. An example of the PTOF sensor with eight tapered regions is shown in Figure 3.

In the next step, the electric arc was excited by electrodes (2) in order to heat up the SMF (1). Then, the holders were moved with defined speed in order to elongate the SMF. After that, the left clamp of the holder was released and the right holder with the SMF was moved left by a distance of 500 µm. The SMF was then fastened back by the left clamp and the electric arc was excited once again. The whole procedure was repeated several times in order to obtain the desired number of waists. To control the geometry of the fabricated PTOF, the electric arc power *P*, travel of the splicer holders *s*, speed of holders *v* and arc duration *t* were precisely defined. The parameters that were set in the fusion splicer during the PTOF fabrication are collected in Table 1. Four different PTOFs were fabricated with two, four, six and eight tapered regions, respectively, as shown in Figure 4. For each type of the sensor, three samples were fabricated. The geometry parameters of the PTOF sensors are collected in Table 2.

## 4. Measurement Setup and Procedure

The measurement setup block diagram is presented in Figure 5a. The fabricated PTOF structures were placed inside a Binder MKT-115 dynamic climate chamber (2), where the temperature *T*_chamber_ was changed from 20 to 70 °C. The temperature increase process was repeated three times for each sample, and during each change the transmission spectrum was analyzed with steps of 1 °C. However, to make the results easier to evaluate, the data are shown with a 5 °C step. The PTOF were connected to the EXFO FTBx-2250 broadband light source (1) and Anritsu MS9740A optical spectrum analyzer (OSA) (3) through the SC/APC pigtails (4), as presented in Figure 5b. The spectrum was investigated in the range from 1460 to 1620 nm in the 501 sampling points. The total power stability of the light source was equal to 0.02 dB.

## 5. Results and Discussion

The presented data are the averages for each type of sensor. The spectral measurement was undertaken via OSA with bandwidth resolution set at 0.44 nm. Spectral characteristics before and after the tapering processes for all PTOF structures are shown in Figure 6. To obtain the transmission spectrum, the difference in optical power between these spectra was calculated. The maximal values of a standard deviation were 0.24, 0.28, 0.59 and 0.45, respectively for PTOF-2, PTOF-4, PTOF-6 and PTOF-8 sensors. Theoretically determined (Equation (3)) transmission dip points are marked on Figure 6 with blue circles. Small discrepancies may have occurred due to imprecise determination of *L*. For PTOF-8 PTOF-6, the calculated and measured FSR (Table 3) are convergent. However, for PTOF-6, a theoretical transmission dip around 1507 nm was not observed. For PTOF-2 and PTOF-4, the measured transmission dips are very shallow and hardly perceptible. A similar observation was made by Yoon et al. [62]. The researchers investigated the temperature and strain sensors based on a micro-tapered fiber grating. They also observed the proportional relation between the number of tapered regions and depth of transmission dip. The explanation of these results is provided in Section 5.1.

The influence of temperature on sensor transmission characteristics for the whole spectrum is presented in Figure 7. The biggest transmission variations caused by temperature change are observed for PTOF-6 and PTOF-8 sensors. Detailed characterization of sensors is given in a later section of this article.

### 5.1. PTOF with Two and Four Waists

The PTOF structures with two waists were investigated as in the case of the first structure. The transmission dip evolution forced by temperature change is given in Figure 8a. Based on the obtained results shown in Figure 8b, it can be stated that, for PTOF-2 sensors, temperature does not affect the transmission spectra in a clear way. The same situation was recorded for PTOF-4 sensors (Figure 9). This can be explained by the fact that the coupling strength between core and cladding modes depends on the number of tapered regions, because they can be considered as grating patterns. A higher number will enhance the coupling [62]; however, for a low number of tapered regions, the coupling strength will be low. In consequence, the depth of transmission dips in PTOF-2 and PTOF-4 is very shallow and it is extremely hard to measure any transmission dip shift. Moreover, the geometry of the tapered regions, e.g., taper-down and taper-up regions that are too smooth, or sensor lengths that are too short, can explain the weak mode coupling. We plan to investigate the sensor geometry in our further research.

### 5.2. PTOF with Six and Eight Waists

More valuable results were observed for PTOF-6 sensors (Figure 10). For the first transmission dip there is a good linear relationship (R^2^ > 0.96) between the transmission variation and temperature, and the sensor resolution reached 0.015 dB/°C. Moreover, the shift of transmission dip was observed and the slope of its linear approximation (R^2^ > 0.97) was equal to 0.39 nm/°C (Figure 10b). Importantly, PTOF-6 provides high temperature sensitivity for the whole temperature range. Excellent temperature sensitivity was recorded for PTOF-8 sensors. The transmission variation can be defined by a linear function (R^2^ ≈ 0.99) with a high (about 4.5 times higher than for PTOF-6) resolution of 0.07 dB/°C (Figure 11). With this sensor, the temperature change can be also determined by an investigation of the transmission dip shift. PTOF-8 obtained a resolution of 0.27 nm/°C, which is about 30% lower than that of the PTOF-6 sensor.

### 5.3. Discussion

The PTOF structures with two and four waists were not recognized as temperature sensors. The number of tapered regions was too few to provide sufficient mode coupling. Therefore, the transmission dips were hardly visible, and their potential change due to temperature variation was not observed. A higher number of tapering regions supported strong mode coupling [62], which resulted in a rise in the depth of transmission dips. Therefore, for PTOF-6 and PTOF-8, the transmission dips were clearly visible. In addition, the larger number of tapered regions, the higher attenuation of the optical signal. In consequence, the number of created waists is limited by the sensitivity of the OSA. For PTOF-6 and PTOF-8, the temperature change affects the transmission shift, in addition to the transmission dip shift. These changes are almost linear in the whole investigated temperature range, which is an excellent feature of the developed sensors. 

As previously mentioned, the fiber optic temperature sensors based on MZI can achieve very high sensitivity of ~2–6.8 nm/°C [44,45,46]. However, these sensors have a very complex construction. In Table 4, we compare our sensors to similar constructions. Our PTOF-6 has at least 4.4 times higher sensitivity in comparison to others, and the operating temperature range is similar to the results found in [7,21,38,59]. Because the temperature range is 20–70 °C, the sensors can be utilized to monitor the temperature of telecommunication infrastructure, e.g., in data centers as a support for existing control systems [63,64]. They can be also employed in numerous indoor applications affected by high electromagnetic interference, temperature monitoring of battery packages during fast charging and discharging [65,66], and hazardous locations. 

## 6. Conclusions

In conclusion, we showed the relationship between the geometry of PTOF structures and temperature sensing ability. The geometry of the simplest PTOF-2 and PTOF-4 sensors did not allow any temperature measurements to be made. Only PTOF-6 and PTOF-8, due to their long interferometric distance and sufficient number of tapered regions, enabled temperature sensing. 

The experimentally determined transmission dips are consistent with the theoretical calculations for PTOF-6 and PTOF-8 sensors. The maximum transmission resolution was equal to 0.07 dB/°C for the PTOF-8 sensor, and the transmission dip shift was equal to 0.39 nm/°C (PTOF-6) and 0.27 nm/°C (PTOF-8). In our future work we plan to develop PTOFs with one or two tapered regions. Then, we will investigate the influence of the geometry of the tapered region (the slope of the tapered-up and tapered-down regions, in addition to the length and diameter of the waists) on the sensing ability in such structures.

## Figures and Tables

**Figure 1 sensors-21-08358-f001:**
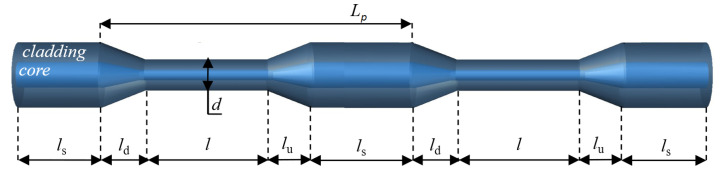
Construction of the PTOF.

**Figure 2 sensors-21-08358-f002:**
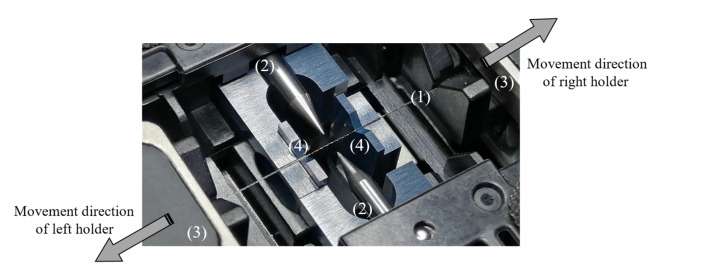
The PTOF structure placed between two electrodes in the fusion splicer.

**Figure 3 sensors-21-08358-f003:**
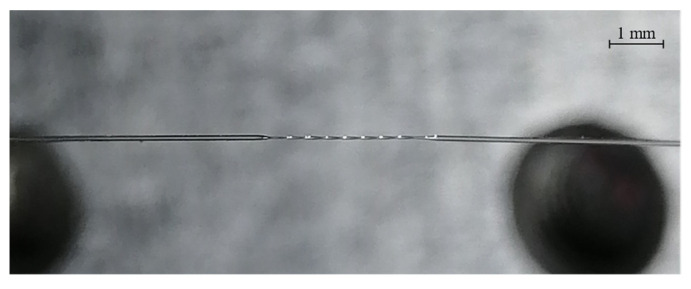
The PTOF with eight waists.

**Figure 4 sensors-21-08358-f004:**
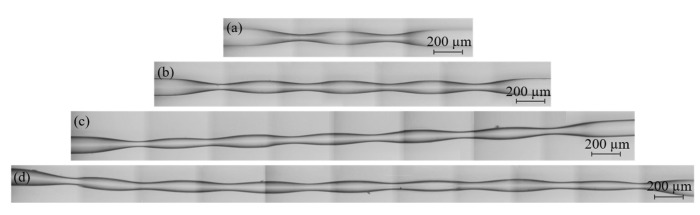
The PTOF with two (**a**), four (**b**), six (**c**), and eight (**d**) tapered regions.

**Figure 5 sensors-21-08358-f005:**
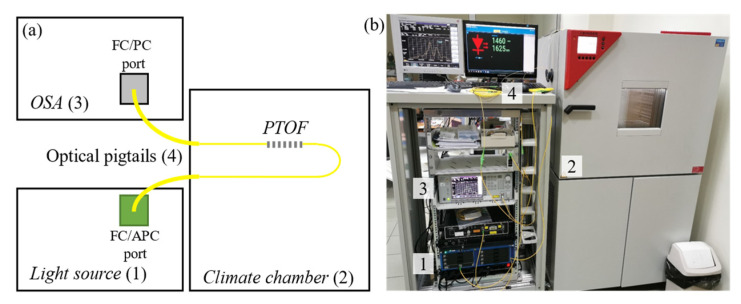
Block diagram of the test bench (**a**) and measurement setup (**b**).

**Figure 6 sensors-21-08358-f006:**
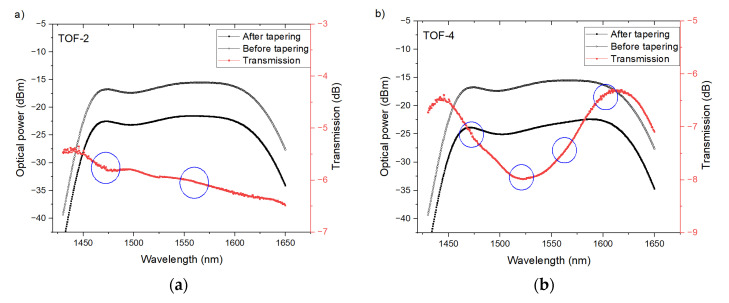
Spectral and transmission characteristics of investigated PTOFs with two (**a**), four (**b**), six (**c**), and eight (**d**) waists.

**Figure 7 sensors-21-08358-f007:**
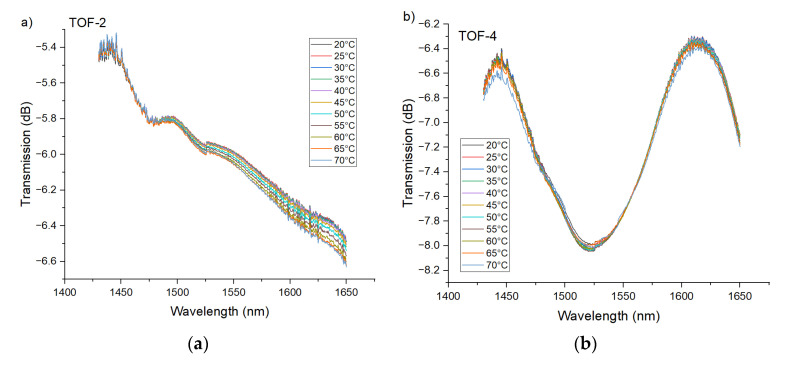
Influence of temperature on transmission characteristics of the PTOF structures with two (**a**), four (**b**), six (**c**), and eight (**d**) waists.

**Figure 8 sensors-21-08358-f008:**
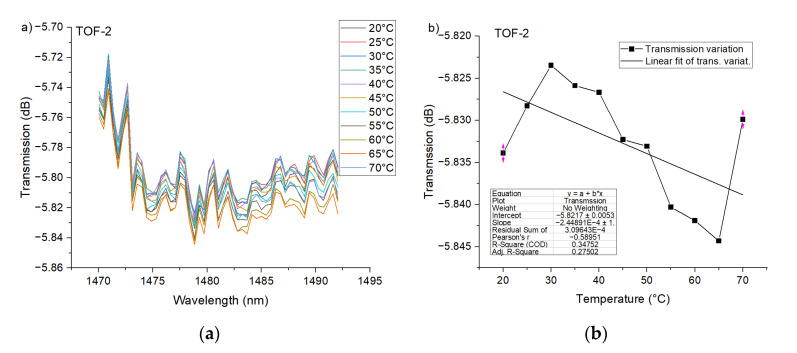
Experimental transmission spectra of the PTOF structure with two waists for the heating process (**a**), and transmission variation as a function of *T*_chamber_ for *λ* = 1475 nm (**b**).

**Figure 9 sensors-21-08358-f009:**
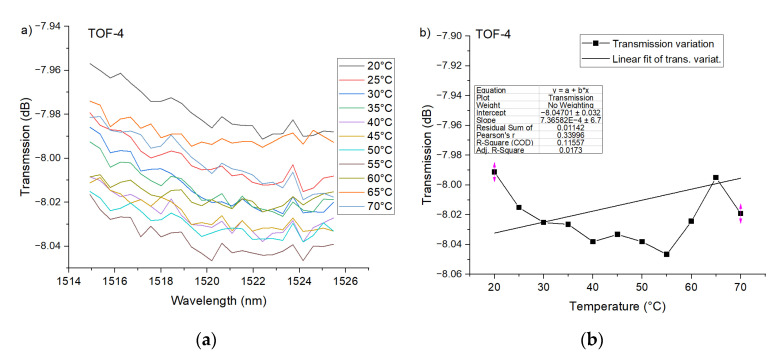
Experimental transmission spectra of the PTOF structure with four waists for the heating process (**a**), and transmission variation as a function of *T*_chamber_ for *λ* = 1520 nm (**b**).

**Figure 10 sensors-21-08358-f010:**
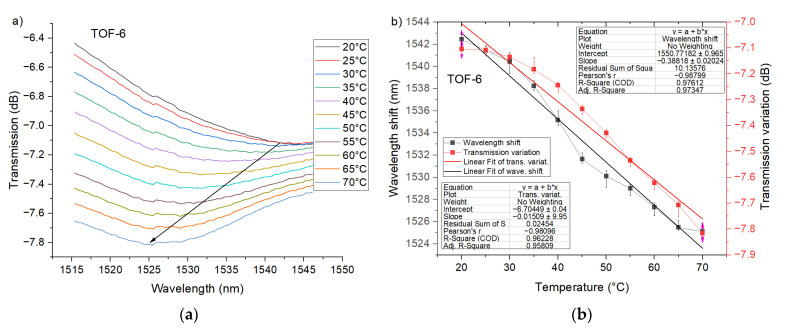
Experimental transmission spectra of PTOF structure with six waists for the heating process (**a**), and transmission variation and wavelength shift as a function of *T*_chamber_ for *λ* = 1545 nm (**b**).

**Figure 11 sensors-21-08358-f011:**
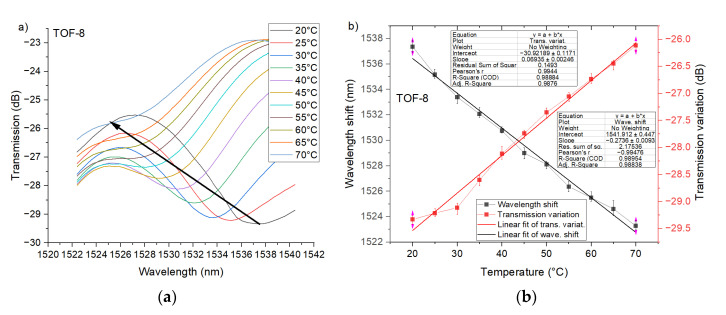
Experimental transmission spectra of the PTOF structure with eight waists for the heating process (**a**), and transmission variation and wavelength shift as a function of *T*_chamber_ for *λ* = 1538 nm (**b**).

**Table 1 sensors-21-08358-t001:** Parameters of the fusion splicer.

Electric Arc Power *P* (mW)	Arc Duration *t* (s)	Holder Travel *s* (µm)	Speed of Holders *v* (µm/s)
20	1.5	500	122

**Table 2 sensors-21-08358-t002:** Parameters of the PTOF structures.

Name	Waist No.	Avg. Waist Diameter *d* (µm)	Min/Max Waist Diameter *d* (µm)	Avg. Waist Period (µm)
PTOF-2	2	44.3	41.4/47.3	620
PTOF-4	4	48.9	46.4/51.6	605
PTOF-6	6	49.7	45.3/53.0	533
PTOF-8	8	51.1	49.0/53.0	581

**Table 3 sensors-21-08358-t003:** FSR parameters’ calculation.

Name	∆*n*_eff_	*L* (mm)	*λ*_dip_ (nm)	Theoretical FSR (nm)	Experimental FSR (nm)
PTOF-2	0.02	1.2	1475	87	85
PTOF-4		2.4	1475	45	49
PTOF-6		3.2	1545	38	44
PTOF-8		4.6	1524	24	22

**Table 4 sensors-21-08358-t004:** Comparison of sensing performance of existing sensors.

Type	Temperature Range	Sensitivity	Ref.
FBG on microfiber	22.5–95 °C	31.32 pm/°C	[21]
Tapered LPG	0–60 °C	87 pm/°C	[7]
Weakly coupled multicore fiber taper	0–1000 °C	36.8 pm/°C	[31]
Tapered few modes fiber with LPG	30–90 °C	39.3 pm/°C	[38]
Tapered microfiber	50–800 °C	13.4 pm/°C	[67]
Abrupt tapered single-mode fiber	15–50 °C	0.0829 dBm/°C	[59]
PTOF SMF	20–70 °C	390 pm/°C and 270 pm/°C	This study

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
