# Peer review of "Temperature Sensor Based on Periodically Tapered Optical Fibers"

_sensors, 2021, doi:10.3390/s21248358_

Round 1

Reviewer 1 Report

In this manuscript, the authors investigate the thermal dependence of the transmission spectrum of different periodically tapered optical fibers (PTOF). They also experimentally demonstrate the feasibility of PTOF as a temperature sensor. The results are reasonable. Therefore, the manuscript might be suitable for the publication on Sensors if the following concerns are properly addressed.

  1. Fiber Fabry-Perot sensor is also an important type of sensor for temperature measurement (see the references below). The author should include those work in the introduction.

1) Lee, Chung E., and Henry F. Taylor. "Fiber-optic Fabry-Perot temperature sensor using a low-coherence light source." Journal of lightwave technology 9.1 (1991): 129-134.

2) Chen, Zhenshi, et al. "High-temperature sensor based on Fabry-Perot interferometer in microfiber tip." Sensors 18.1 (2018): 202.

3) Chen, Mao-qing, et al. "High sensitivity temperature sensor based on fiber air-microbubble Fabry-Perot interferometer with PDMS-filled hollow-core fiber." Sensors and Actuators A: Physical 275 (2018): 60-66.

  1. Starting from line 31, the author mentioned that multicore fibers has been proposed for high temperature measurement up to 1000 C. There are also sapphire-fiber-based sensors that can measure much higher temperature (see the references below). The author should include those work in the introduction.

1). Yang, Shuo, et al. "All-sapphire miniature optical fiber tip sensor for high temperature measurement." Journal of Lightwave Technology 38.7 (2019): 1988-1997.

2). Grobnic, Dan, et al. "Sapphire fiber Bragg grating sensor made using femtosecond laser radiation for ultrahigh temperature applications." IEEE Photonics Technology Letters 16.11 (2004): 2505-2507.

3). Yang, Shuo, Di Hu, and Anbo Wang. "Point-by-point fabrication and characterization of sapphire fiber Bragg gratings." Optics letters 42.20 (2017): 4219-4222.

4). Habisreuther, Tobias, et al. "Sapphire fiber Bragg gratings for high temperature and dynamic temperature diagnostics." Applied Thermal Engineering 91 (2015): 860-865.

5). Mihailov, Stephen J., Dan Grobnic, and Christopher W. Smelser. "High-temperature multiparameter sensor based on sapphire fiber Bragg gratings." Optics letters 35.16 (2010): 2810-2812.

  1. In Line 43, the meaning of the abbreviation PTOF is not defined in the main body of the manuscript.

  1. How did the authors quantify the transmission variation in Fig.8 and 9?

Reviewer 2 Report

Present work is on PTOF based temperature sensors in the range from 20-70 C. Mainly the geometry of the sensors have been investigated. The work could be of interest to scientific community working in this area. However, the presentation of the work is ambiguous and needs major revision. The work may be considered for publication after inclusion of following comments: 1. The language is very poor and needs betterment. 2. Pp1. Line 36—The sensitivity should be an absolute value. Moreover, be consistent with units. 3. Pp1, line 40—The knowledge gap is not explicitly stated. Please add more relevant literature and hence knowledge gap. 4. Pp4, figure 6.. Authors stated that 3 samples of each type were prepared. Can you please add standard deviation/error in the graph? 5. Pp6, line 135 -- An understandable explanation for PTOF-2 and 4 sensors towards their temperature versus transmission relation is missing. Please elaborate your opinions. 6. Pp6, line 149-- Authors claim that PTOF-6 provides high temperature sensitivity. Can you please compare its sensitivity with the existing same category sensors? 7. In the results and discussion part, only the “results” part is showed. However the reviewer couldn’t understand the discussion part. There is lack of physics/knowledge on why PTOF-2 and 4 showed poor performance. Authors are encouraged to add a nice discussion part. 8. It would be better to add some applications of the present sensor. 9. The legends and font size of all figures may be enlarged.

Reviewer 3 Report

The manuscript reported a temperature sensor based on periodically tapered optical fibers. The theory was analyzed and the sensor was fabricated and tested. Sensitivities of 0.07 dB/C and 0.39 nm/C were reached. However, the performance of the sensors that have 6+ periods align with the theoretical analysis. I believe a lot of further experiments are needed to make this work complete.

  1. How was the temperature range of 20-70 C decided for the sensor?
  2. In section 5.1, the mismatch between the calculated and measured performance was only briefly speculated, and the problems were not addressed. If the problem is too smooth tapers, please present measurements with tapers that have different smoothness.
  3. The error bars need to be added to the results for PTOF with six and eight waists.
  4. According to the results, it seems that the better waists there are, the better the performance will be. In this case, how about PTOF with ten waists? Will it have better performance than that of PTOF-8? If so, there will be a better sensor. If not, what could be the disturbing factor? Of course, if the performance is not enhanced significantly but requires much more effort, PTOF-8 will be a good option.

Round 2

Reviewer 2 Report

Authors have provided sufficient responses to the raised comments in the revised work. I recommend acceptance of the work after inclusion of the following point. 

  1. Authors are encouraged to write some lines on why temperature sensors are required rather than directly jump to the fibre optic sensors. Conventional temperature sensors e.g., diode-based, may be included in introduction section. Below given work may be refrerred: 

Vibhor Kumar, Jyoti Verma, A.S.Maan, Jamil Akhtar "Epitaxial 4H–SiC based Schottky diode temperature sensors in ultra-low current range" vol 182, 109590, 2020. https://doi.org/10.1016/j.vacuum.2020.109590

Vibhor Kumar, A. S. Maan, Jamil Akhtar, "Barrier height inhomogeneities induced anomaly in thermal sensitivity of Ni/4H-SiC Schottky diode temperature sensor" Journal of Vacuum Science & Technology B 32, 041203 (2014); https://doi.org/10.1116/1.4884756

Reviewer 3 Report

The manuscript is improved with the added discussion and explanation. I had a hard time finding the direct proof of the relationship between taper numbers and coupling strength in Ref [59] (lines 202 and 146). It turned out that it should be Ref [58]. I recommend the manuscript be published on Sensors after the authors carefully review the numbering of the references.  
